# SOLQ: Segmenting Objects by Learning Queries

**Bin Dong** [*]    **Fangao Zeng** [*]    **Tiancai Wang** [*]    **Xiangyu Zhang**    **Yichen Wei**

MEGVII Technology

{dongbin,zengfangao,wangtiancai,zhangxiangyu,weiyichen}@megvii.com

## Abstract

In this paper, we propose an end-to-end framework for instance segmentation. Based on the recently introduced DETR [1], our method, termed SOLQ, segments objects by learning unified queries. In SOLQ, each query represents one object and has multiple representations: class, location and mask. The object queries learned perform classification, box regression and mask encoding simultaneously in an unified vector form. During training phase, the mask vectors encoded are supervised by the compression coding of raw spatial masks. In inference time, mask vectors produced can be directly transformed to spatial masks by the inverse process of compression coding. Experimental results show that SOLQ can achieve state-of-the-art performance, surpassing most of existing approaches. Moreover, the joint learning of unified query representation can greatly improve the detection performance of DETR. We hope our SOLQ can serve as a strong baseline for the Transformer-based instance segmentation. Code is available at https://github.com/megvii-research/SOLQ.

## 1   Introduction

Instance segmentation, serving as one of visual detection tasks, not only locates instances of different categories but also generates pixel-level mask for each instance. State-of-the-art instance segmentation methods [2, 3, 4, 5] follow the two-stage paradigm, which first performs object detection and then segments the masks within detected boxes by RoIAlign [2]. Those methods are relatively easy to be optimized thanks to the deployment of mature object detectors [6, 7]. However, the segmentation branch heavily relies on the detection branch, making it hard to achieve better joint learning of multiple tasks. Some recent works [8, 9, 10, 11] build instance segmentation frameworks on top of anchor-free object detectors [12, 13] to remove the ROI-Cropping operation, reducing the effect of feature misalignment. For example, CondInst [9] based on FCOS [12] employs dynamic convolutions [14] to perform instance segmentation. YOLACT [8] models the instance segmentation as a combination of prototypes weighted by learned mask coefficients for each anchor. However, the weights of dynamic convolutions or the mask coefficients are still generated by instance proposals.

Regardless of the bounding boxes, SOLO [15] introduces the notion of "instance categories" and segments objects by locations. SOLOv2 [16] further solves the problem of inefficient mask representation by introducing dynamic convolution and so-called matrix Non-Maximum Suppression (NMS). Though SOLO directly outputs instance masks based on locations, hand-crafted post-processes, like NMS, are still required to remove duplicated predictions. Also, the performance of small objects is far from satisfactory due to the imbalance of location samples between the large and small objects. Building an end-to-end instance segmentation framework is still a remaining problem.

Our work is inspired by DETR [1], which first proposes the end-to-end solution for object detection. In DETR, object detection is regarded as a set prediction problem and objects are represented by

---

[*]Equal contribution. This work is supported by The National Key Research and Development Program of China (No.2017YFA0700800) and Beijing Academy of Artificial Intelligence (BAAI).

35th Conference on Neural Information Processing Systems (NeurIPS 2021).

learnable query embeddings. How to encode the spatial binary mask into such an end-to-end system is an opening question. As shown in Fig. 1(a), DETR is further extended to panoptic segmentation by directly reshaping the learnable embeddings into spatial domain and building a FPN-style [17] network to produce the final mask predictions. However, both the Transformer encoder and decoder fail to model the spatial information well. Therefore, it is inappropriate to generate the spatial mask based on such query embeddings. Besides, the spatial mask labels used for supervision are of large resolutions, which results in high computation cost and makes it separated to learn the detection and mask branches[2]. So we need to find one mask representation that satisfies the following conditions: 1) The representation can naturally convert object mask from spatial domain to embedding domain; 2) The process that encodes the spatial masks into embeddings should be reversible; 3) Mask embeddings encoded can keep the principle components of spatial mask. We turn to methods in literature for help and surprisingly find that classical compression coding methods (*e.g.* Sparse Coding [18]) just satisfy these conditions mentioned above. The mask embeddings can be simply generated from the learnable queries. In training phase, ground-truth spatial mask of each instance can be projected into low-dimensional mask embedding by compression coding and the mask embeddings are used to supervise the learning of predicted mask embeddings. In inference phase, binary spatial mask can be reconstructed from predicted mask embedding by the inverse process of compression coding.

With these analysis, we explore how to better encode the spatial mask into the end-to-end object detectors in this paper. Based on DETR, our proposed method, termed SOLQ, segments objects by learning queries. In SOLQ, we formulate the instance segmentation as the joint learning of unified query representation (UQR). The UQR learned can be used to perform parallel predictions for three sub-tasks (classification, localization and segmentation) simultaneously and all predictions are obtained in a regression manner (see Fig. 1(b)). In this way, SOLQ directly outputs instance masks together with corresponding class confidences and box coordinates. The learning of UQR can be divided into two parts: generating instance-aware query embeddings and joint supervision of multi-task learning. Specifically, all candidate instances are initialized with several learnable queries, which interact with extracted image features in Transformer decoder to produce the instance-aware query embeddings. The instance-aware query embeddings are further input to three branches of sub-tasks, which contains several linear projection layers, to generate three sub-task vectors. For the classification and regression branches, we follow the same supervisions as in DETR [1]. For the mask branch, we conduct implicit supervision with the help of mask compression coding mentioned above.

To summarize, our contributions are:

- We propose an end-to-end framework for instance segmentation based on DETR. SOLQ formulates the instance segmentation as the joint learning of UQR. In UQR, the mask representation can be converted from spatial domain into embedding domain, which is consistent with the learnable query embeddings in DETR.

- Experiments show that SOLQ with ResNet101 achieves 40.9% mask AP and 48.7% box AP on the challenging MS COCO dataset [19] without bells and whistles, outperforming SOLOv2 by 1.2% mask AP and 6.1% box AP. It is worthy noting that SOLQ can improve 2.0% in box AP compared to DETR thanks to the joint learning of UQR.

## 2  Related Work

**Instance Segmentation** Instance segmentation is a classic but challenging computer vision task. It is required to output each object instance in image with instance-level category label and localization, along with pixel-level mask simultaneously. Currently, there are mainly three categories of instance segmentation methods: top-down, bottom-up and directly-predict methods. Top-down approaches [20, 2, 8, 21, 22, 23, 3, 5, 24] follow the detect-then-segment pipeline. They first generate bounding boxes by object detectors and segment the masks by ROIAlign [2] or dynamic convolutions [9]. Bottom-up methods [25, 26, 27, 28] learn per-pixel embeddings via semantic label and then cluster them into instance groups. For directly-predict methods, PolarMask [29] employs polar coordinates to represent mask contours. Latest SOLO [15] and SOLOv2 [16] directly segment the objects by locations without dependence on bounding boxes or embedding learning. QueryInst [24] and ISTR [30] extend the Sparse RCNN [31] to perform end-to-end instance segmentation. In this paper, we

---

[2]Train the object detector first and then freeze the weights of object detector to train segmentation branch.

explore an end-to-end instance segmentation solution by learning an unified query representation without any post-processing procedures, like Non-Maximal Suppression (NMS).

**Transformer in Vision** Transformer [32] introduces the self-attention mechanism to model long-range dependencies, and has been widely applied in natural language processing (NLP). Recently, several works attempted to involve the Transformer architecture into various computer vision tasks and showed promising performances. The non-local block [33] is first proposed to enhance video recognition by aggregating spatial information. After that, CCNet [34] further extends the self-attention via sparse attention in semantic segmentation. DETR [1] and Deformable DETR [35] adopt learnable queries and Transformer architecture together with bipartite matching to perform object detection in end-to-end fashion, without any hand-crafted process such as NMS. IPT [36] proposes a transformer-based pretrained network for low-level image processing. ViT series [37, 38, 39, 40] take an image as a sequence of patches and achieve the cross-patch interactions by Transformer architecture in image classification.

**Compression Coding in Vision** Consider the advantage of low-dimension representation and less computation cost, some recent works have attempted to introduce compression coding methods, like Sparse Coding [41], Principal Component Analysis (PCA) [42] and Discrete Cosine Transform (DCT) [43], into computer vision field. For image classification, [44] takes the DCT coefficients obtained from RGB images as the inputs of convolutional neural networks (CNNs) to reduce the communication bandwidth between CPU and GPU. DCT-Mask [45], based on Mask R-CNN, employs DCT supervision to produce high-quality mask representation. Analogously, [46] performed semantic segmentation on the DCT representation and fed the rearranged DCT coefficients to CNNs. MEInst [21] and ISTR [30] encode binary masks into fixed-dimensional mask vectors produced by PCA.

## 3 Method

### 3.1 Reviewing DETR

Recently, DETR [1] succeed in object detection. It formulates object detection as a set prediction problem and introduces object queries, a set of learnable embeddings, to represent objects. In DETR, each object query predicts an object for a given input image and set prediction loss is adopted to achieve one-to-one matching between the predicted and ground-truth objects in training phase. Further, the transformer encoder-decoder architecture is employed to model the relation between query embeddings and instances for better one-to-one set prediction.

**Set Prediction Loss** Let $y = (c, b)$ and $\hat{y} = (\hat{c}, \hat{b})$ denote the ground truths $y$ and the set of predictions $\hat{y}$, respectively. $c, \hat{c} \in \mathbb{R}^{J \times S}$ are the corresponding class labels and predicted class scores, where $J$ and $S$ are the object number and class number. $b, \hat{b} \in \mathbb{R}^{J \times 4}$ are the corresponding ground-truth and predicted box coordinates. $\omega \in \Omega_J$ is the assignment between the ground truths and predictions. Then the optimal one-to-one assignment $\omega^*$ can be calculated by bipartite matching [47] as:

$$\omega^* = \underset{\omega \in \Omega_J}{arg\min} \mathcal{L}^{det}(y, \omega(\hat{y})) \tag{1}$$

where the bipartite matching loss for object detection $\mathcal{L}_{det}$ can be summarised as:

$$\mathcal{L}_{det}(y, \hat{y}) = \lambda_{cls} \cdot \mathcal{L}_{cls}(c, \hat{c}) + \lambda_{L_1} \cdot \mathcal{L}_{L_1}(b, \hat{b}) + \lambda_{giou} \cdot \mathcal{L}_{giou}(b, \hat{b}) \tag{2}$$

Here $\mathcal{L}_{cls}$ denotes the focal loss [48] for classifications, $\mathcal{L}_{L_1}$ and $\mathcal{L}_{giou}$ are L1 loss and generalized IoU loss [49] for box coordinates, respectively. $\lambda_{cls}$, $\lambda_{L_1}$ and $\lambda_{giou}$ are corresponding coefficients.

**Extension on Segmentation** As shown in Fig. 1(a), DETR is further generalized to panoptic segmentation task by adding a multi-head attention (MHA) and FPN-style CNN after the Transformer decoder. Features from the encoder and learned query embeddings from the decoder are reshaped to spatial domain and then interact in MHA. The produced features are then gradually upsampled to the image size by the FPN-Style CNN to obtain spatial masks. It's easy to adapt this framework to perform instance segmentation by cropping instance masks within detected bounding boxes.

**Object Representation** In DETR, objects are represented as a set of object queries. Object queries are initialized by the learnable query embeddings and then interact with the image features in the transformer decoder to update their representation. Finally, object query, as an unified representation, is directly used to classify and localize objects. However, the representation of spatial mask is built by

compulsorily reshaping the query embeddings to spatial domain in instance segmentation task. Such design leads to different representation forms compared to detection branch. Besides, the two-stage training process makes DETR fail to enjoy the benefit from multi-task learning.

## 3.2 Network Architecture

To encode the spatial mask into the end-to-end object detector in an unified form, we present a simple but efficient framework for instance segmentation based on DETR. The unified query representation (UQR) is proposed to perform instance-level localization and pixel-level segmentation simultaneously. The compression coding is further introduced to project the spatial mask into embedding domain for high-quality and efficient mask representation. The overall architecture of SOLQ is showed in Fig. 1(b). SOLQ can be divided into three parts: feature extraction network, Transformer decoder and unified query representation. We will describe our method in detail as follows.

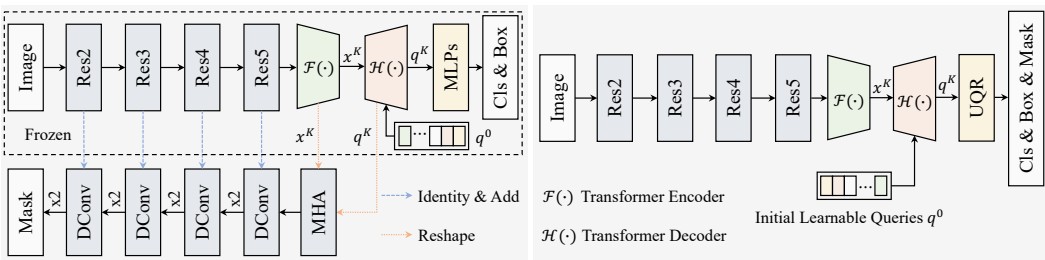

(a) DETR FPN-style instance segmentation      (b) SOLQ directly-predict style instance segmentation

Figure 1: Architecture comparison between the naive DETR and the proposed SOLQ for instance segmentation. "Res2-Res5" are different stages of ResNet [50]. "DConv" and "MHA" are deconvolution [51] layer and multi-head attention, respectively. "MLPs" means multi-layer perceptions and "UQR" denotes unified query representation, which is introduced in detail via Fig. 2. $q^0$ is initial learnable object queries. "Frozen" means that train object detector firstly, and then freeze weights of the object detector to train instance segmentation branch, separately.

**Feature Extraction Network** The feature extraction network consists of the backbone and Transformer encoder. Given an image $I \in \mathbb{R}^{H \times W \times 3}$, ResNet [50] is used as the backbone to extract basic feature map $x^0 \in \mathbb{R}^{C \times \frac{H}{32} \times \frac{W}{32}}$, where $H$, $W$, $C$ are the height, width and channels of feature map, respectively. Then the basis feature $x^0$ is fed into $K$ Transformer encoder layers to get the refined feature map $x^K \in \mathbb{R}^{C \times \frac{HW}{32^2}}$ via $\{x^k = \mathcal{F}^k(x^{k-1})\}_{k=1}^K$, iteratively. Each Transformer encoder layer $\mathcal{F}^k(\cdot)$ is composed of a multi-head self-attention (MHSA) and a feed-forward network (FFN).

**Transformer Decoder** Given the learnable object queries, we generate the instance-aware query embeddings for the unified query representation by the Transformer decoder. In details, a set of learnable object queries $q^0 \in \mathbb{R}^{J \times C}$ are firstly randomly initialized. Then the initial object queries $q^0$ interact with the refined feature map $x^K$ in $K$ Transformer decoder layers to obtain instance-aware query embeddings $q^K \in \mathbb{R}^{J \times C}$ by $\{q^k = \mathcal{H}^k(q^{k-1}, x^K)\}_{k=1}^K$. Each Transformer decoder layer $\mathcal{H}^k(\cdot)$ has an extra multi-head cross-attention layer compared to the Transformer encoder layer. The instance-aware query embeddings $q^K$ are then fed into unified query representation part to generate predictions for three sub-tasks, including classification, localization and segmentation.

**Unified Query Representation** After Transformer decoder, each instance-aware query embedding in $q^K$ represents the features of corresponding instance. The supervision of three sub-tasks (classification, localization and segmentation) in an unified form (*e.g.* vector) is the last piece of the puzzle to achieve parallel predictions. Fig. 2 shows the learning of UQR. We mainly describes the joint supervision of multi-task learning as well as the training and inference processes of mask branch.

In details, UQR is learned under the supervision of classification, localization and mask branches. Both classification and localization branches are the same as in DETR [1]. The classification branch is a fully-connected (FC) layer that predicts the class confidences $\hat{c}$. The localization branch is a multi-layer perception (MLP) with hidden size 256 and predicts 4 box coordinates $\hat{b}$. Similar to localization branch, the mask branch is also a multi-layer perception with hidden size 1024 and predicts mask vectors $\hat{v} \in \mathbb{R}^{J \times n_k}$, $n_k$ is the dimension of each mask vector. During training, the mask

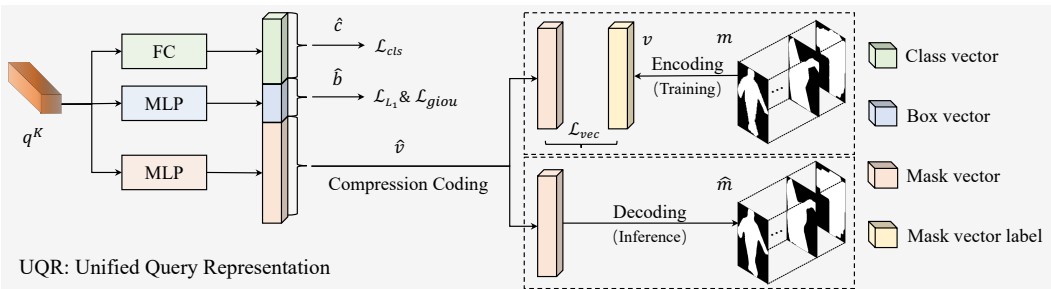

Figure 2: The learning of the proposed unified query representation. $q^K$ refers to the learned instance-aware query embeddings. $\hat{c}$, $\hat{b}$, $\hat{v}$ are predicted class, box and mask vectors, respectively. $\hat{m}$ is the binary masks reconstructed. $m$ and $v$ denote ground-truth binary masks and mask vectors, correspondingly. $\mathcal{L}_{cls}$, $\mathcal{L}_{L_1}\&\mathcal{L}_{giou}$ and $\mathcal{L}_{vec}$ are the losses for classification, box regression and mask segmentation, respectively.

vectors predicted are supervised by the ground-truth mask vectors $v \in \mathbb{R}^{J \times n_k}$ generated from spatial mask $m \in \mathbb{R}^{J \times N \times N}$ by the mask compression coding described below, $N$ is the spatial dimension of binary mask. While for the inference, the predicted mask vectors $\hat{v}$ can be used to reconstruct the spatial masks $\hat{m}$ by the inverse process of compression coding. Note that each Transformer decoder layer learns such an UQR and auxiliary supervision is adopted for better performance.

**Mask Compression Coding** As mentioned above, predicted mask vectors $\hat{v}$ generated by the mask branch are supervised by the ground-truth mask vectors $v$. Here, we explore three compression coding methods to transform 2D spatial binary masks into 1D mask vectors, including Sparse Coding [41], Principal Component Analysis (PCA) [42] and Discrete Cosine Transform (DCT) [43].

*Sparse Coding* compresses the binary mask as a sparse combination of $n_k$ atoms from an overcomplete dictionary $D \in \mathbb{R}^{n_k \times N^2}$. Ground-truth mask vectors $v$ can be obtained from $m$ through solving the minimum of Lasso [52] problem:

$$(v^*, D^*) = \underset{(v,D)}{arg\min}(\frac{1}{2}||m - vD||_2^2 + \beta||v||_1), \ s.t. \ ||D_e||_2 = 1, \ \forall e \in [1, n_k] \tag{3}$$

where $\beta$ is the regular coefficient. The binary masks $\hat{m}$ can be reconstructed via $\hat{m} = \hat{v}D$.

*PCA* transforms binary masks to low-dimensional mask vectors via matrix factorization. The process can be summarized as the following optimization problem:

$$P^* = \underset{P}{arg\min}||m - mPP^T||^2, \ s.t. \ PP^T = U_{n_k} \tag{4}$$

where $P \in \mathbb{R}^{N^2 \times n_k}$ and $U_{n_k} \in \mathbb{R}^{n_k \times n_k}$ are the projection matrix and unit matrix, respectively. Ground-truth mask vectors $v$ can be represented as $v = mP$ meanwhile binary masks can be reconstructed by $\hat{m} = \hat{v}P^T$.

*DCT* first transforms ground-truth binary masks $m$ into frequency domain according to $f = AmA^T$, where $A \in \mathbb{R}^{N \times N}$ is the transform matrix and the $(h, l)$ element of A can be calculated by $A_{h,l} = \sqrt{\frac{1 + sign(h)}{N}}cos[\frac{(l+0.5)\pi}{N}h]|$. The ground-truth mask vectors $v$ can be encoded by sampling the low-frequency components from $f$. The binary masks $\hat{m}$ can be recovered through the inverse sampling and transformation $\hat{m} = A^{-1}\hat{f}(A^T)^{-1}$, where $\hat{f}$ is the inverse sampling result from $\hat{v}$.

**Loss Function** We extend object ground-truths with mask vectors as $y = (c, b, v)$ and corresponding predictions are $\hat{y} = (\hat{c}, \hat{b}, \hat{v})$. The overall loss function for supervision can be expressed as:

$$\mathcal{L}_{inst} = \mathcal{L}_{det} + \lambda_{vec} \cdot \mathcal{L}_{vec}(v, \hat{v}) \tag{5}$$

where $\mathcal{L}_{vec}$ is the mask vector loss and we use L1 loss in practice. $\lambda_{vec}$ is the corresponding weight and $\mathcal{L}_{det}$ is same as Eq. 2. Note that mask loss is not included in bipartite matching. The participation of mask loss may affect the global matching between the object queries and ground-truths.

### 3.3 Discussion

Whereas our SOLQ shares similarities with MEInst [21], DCT-Mask [45] and ISTR [30] in the compression coding of spatial mask, the main differences are described as follows. SOLQ produces the instance masks together with the instance class and the location in a parallel way. It aims to learn an unified query representation for better multi-task learning. The predictions of three sub-tasks are all obtained in a regression manner. Also, SOLQ is an end-to-end instance segmentation framework without any post-processes, like NMS. In contrast, DCT-Mask encodes the instance masks based on the RoI features cropped by the bounding boxes from detection branch. The mask encoding in MEInst and ISTR requires extra optimization process to obtain the optimal project matrix for reconstructing spatial masks. So they are not learned in an end-to-end manner. Also, our concurrent work, QueryInst [24] performs instance segmentation in an end-to-end fashion based on Sparse RCNN [31]. The overall architecture mainly follows 'detect-then-segment' paradigm. The features cropped by bounding boxes interact with the updated queries to generate the segmentation masks.

## 4 Experiments

### 4.1 Dataset and Metrics

We validate our method on COCO benchmark [19]. COCO contains 115k images for training, 5k for validation and 20k for testing, involving 80 object categories with instance-level segmentation annotations. We report results on COCO 2017 *test-dev* set for state-of-the-art comparison and the results on COCO 2017 *val* set for ablation studies. Consistent with Mask R-CNN [2], the standard COCO metrics including $AP^{box}$, $AP^{box}_S$, $AP^{box}_M$, $AP^{box}_L$, and $AP^{seg}$, $AP^{seg}_S$, $AP^{seg}_M$, $AP^{seg}_L$ are used to evaluate the performance of object detection and segmentation.

### 4.2 Implementation Details

For fast convergence, we build the SOLQ on top of Deformable DETR [35] in practice. ResNet [53], pretrained on the ImageNet [54] is employed as the backbone and multi-scale feature maps from C3 to C6 stages are used. For the deformable attention, the number of heads is set as 8 and the number of sampling points is set as 4. For the mask branch, $n_k$ is set to 256, hidden dim of MLP is 1024 and $\lambda_{vec}$ is 3.0. Following DETR, $\lambda_{cls} = 2$, $\lambda_{L1} = 5$, $\lambda_{giou} = 2$. We train our model with Adam optimizer with weight decay of $1.0 \times 10^{-4}$. Models are trained for 50 epochs with the initial learning rate $2.0 \times 10^{-4}$ and decayed at $40^{th}$ epoch by a factor 0.1. Multi-scale training is adopted, where the shorter side is randomly chosen within [408, 800] and the longer side is less or equal to 1333. All experiments are conducted over 8 Tesla V100 GPUs with batch size 32 except the comparison in Sec. 4.4. Since the D-DETR with SQR can only be trained with batch size 16, we also perform the D-DETR with UQR under the same setting for fair comparison.

### 4.3 Comparison with State-of-the-arts

As shown in Tab. 1, we compare SOLQ with state-of-the-art methods on COCO *test-dev* set. Our method achieves best performance on both $AP^{seg}$ and $AP^{box}$ metrics. Compared to the typical two-stage methods Mask R-CNN [2] and Cascade Mask R-CNN [7], SOLQ with ResNet-101 surpasses in $AP^{seg}$ by 2.1% and 0.9%, respectively. Besides, we also compare SOLQ with state-of-the-art one-stage methods CondInst [9] and SOLOv2 [16], which are built based on dynamic convolution. Our method outperforms them 1.8% and 1.2% in $AP^{seg}$, respectively. Further, based on the recently introduced Swin Transformer [40], our SOLQ will serve as a fully-Transformer framework for instance segmentation and it can achieve 46.7% $AP^{seg}$ and 56.5% $AP^{box}$. To further validate the quality of boundary prediction, we also evaluate SOLQ using the Boundary AP [55] in Appendix A.1.

It is worthy noting that SOLQ performs well on objects of different scales, especially on small and middle scale objects. For example, SOLQ with ResNet-101 surpasses SOLOv2 by 5.2% in $AP^{seg}_S$ and 0.9% in $AP^{seg}_M$, respectively. It should be owing to the mask compression encoding of spatial binary mask. In SOLOv2, the mask predictions are supervised by the ground-truth mask of 1/4 instance size so the performance of small objects are not optimized well. For our SOLQ, the mask

Table 1: State-of-the-art comparison on the COCO 2017 *test-dev* set. All the models are trained with multi-scale and tested with single scale. '†' and '*' are the results reported in ISTR [24] and QueryInst [24], respectively. For experiments with Swin-L backbone [40], the shorter side of input is randomly chosen within [400, 1200] and the longer side is less or equal to 1536.

| Method | Backbone | Epochs | $AP^{seg}$ | $AP^{seg}_S$ | $AP^{seg}_M$ | $AP^{seg}_L$ | $AP^{box}$ | $AP^{box}_S$ | $AP^{box}_M$ | $AP^{box}_L$ |
|---|---|---|---|---|---|---|---|---|---|---|
| Mask R-CNN† [2] | R50-FPN | 36 | 37.5 | 21.1 | 39.6 | 48.3 | 41.3 | 24.2 | 43.6 | 51.7 |
| Cascade Mask R-CNN* [7] | R50-FPN | 36 | 38.6 | 21.7 | 40.8 | 49.6 | 44.5 | - | - | - |
| HTC* [4] | R50-FPN | 36 | 39.7 | 22.6 | 42.2 | 50.6 | 44.9 | - | - | - |
| MEInst† [21] | R50-FPN | 36 | 33.5 | 19.3 | 35.7 | 42.1 | 42.5 | 25.6 | 45.1 | 52.2 |
| CondInst [9] | R50-FPN | 36 | 37.8 | 21.0 | 40.3 | 48.7 | 42.1 | 25.1 | 44.5 | 52.1 |
| BlendMask [22] | R50-FPN | 36 | 37.8 | 18.8 | 40.9 | 53.6 | 43.0 | 25.3 | 45.4 | 54.0 |
| SOLOv2 [16] | R50-FPN | 72 | 38.8 | 16.5 | 41.7 | 56.2 | 40.4 | 20.5 | 44.2 | 53.9 |
| QueryInst [24] | R50-FPN | 36 | 40.6 | 23.4 | 42.5 | 52.8 | 45.6 | - | - | - |
| ISTR [30] | R50-FPN | 36 | 38.6 | 22.1 | 40.4 | 50.6 | 46.8 | 27.8 | 48.7 | 59.9 |
| **SOLQ**, *ours* | R50 | 50 | 39.7 | 21.5 | 42.5 | 53.1 | 47.8 | 27.6 | 50.9 | 61.6 |
| Mask R-CNN† [2] | R101-FPN | 36 | 38.8 | 21.8 | 41.4 | 50.5 | 43.1 | 25.1 | 46.0 | 54.3 |
| Cascade Mask R-CNN* [7] | R101-FPN | 36 | 40.0 | 22.5 | 42.5 | 51.2 | 46.2 | - | - | - |
| HTC* [4] | R101-FPN | 36 | 40.8 | 23.0 | 43.5 | 52.6 | 46.3 | - | - | - |
| MEInst† [21] | R101-FPN | 36 | 35.3 | 20.4 | 37.8 | 44.5 | 44.5 | 26.8 | 47.3 | 54.9 |
| CondInst [9] | R101-FPN | 36 | 39.1 | 21.5 | 41.7 | 50.9 | 43.5 | 25.8 | 46.0 | 54.1 |
| BlendMask [22] | R101-FPN | 36 | 39.6 | 22.4 | 42.2 | 51.4 | 44.7 | 26.6 | 47.5 | 55.6 |
| DCT-Mask [45] | R101-FPN | 36 | 40.1 | 22.7 | 42.7 | 51.8 | - | - | - | - |
| SOLOv2 [16] | R101-FPN | 72 | 39.7 | 17.3 | 42.9 | 57.4 | 42.6 | 22.3 | 46.7 | 56.3 |
| QueryInst [24] | R101-FPN | 36 | 42.8 | 24.6 | 45.0 | 55.5 | 48.1 | - | - | - |
| ISTR [30] | R101-FPN | 36 | 39.9 | 22.8 | 41.9 | 52.3 | 48.1 | 28.7 | 50.4 | 61.5 |
| **SOLQ**, *ours* | R101 | 50 | 40.9 | 22.5 | 43.8 | 54.6 | 48.7 | 28.6 | 51.7 | 63.1 |
| QueryInst [24] | Swin-L | 50 | 49.1 | 31.5 | 51.8 | 63.2 | 56.1 | - | - | - |
| **SOLQ**, *ours* | Swin-L | 50 | 46.7 | 29.2 | 50.1 | 60.9 | 56.5 | 37.6 | 60.0 | 70.6 |

compression encoding encodes the high-resolution binary mask (e.g. $128 \times 128$) into low-dimension mask vectors using the sparsity characteristic of the binary mask and keeps the principle information.

## 4.4 UQR *vs.* SQR

In this section, we show the comparison between the separate query representation (SQR) (see Fig. 1(a)) in DETR and the UQR used in SOLQ (see Fig. 1(b)). As shown in Tab. 2, we surprisingly find that UQR in SOLQ can boost the detection performance of DETR with a great margin, improving $AP^{box}$ by 2.3% and 2.0% with ResNet50 and ResNet101. While in comparison, separate query representation (SQR) only improve the $AP^{box}$ by 0.1% and the segmentation performance with 33.4 $AP^{seg}$ is much lower than that of UQR. The large improvement in $AP^{box}$ shows the effectiveness of our proposed UQR, owing to the unified learning of query representation. For efficiency comparison between them, please refer to the Appendix A.2.

We also report the detection performance of both Faster R-CNN and Mask R-CNN since Mask R-CNN is built on top of Faster R-CNN. Compared to Faster R-CNN, Mask R-CNN improves $AP^{box}$ by 0.6% and 1.1% with ResNet50-FPN and ResNet101-FPN, respectively. As we see, multi-task learning tends to improve the performance with each other and the performance can be further improved if these sub-tasks are learned using an unified representation. SOLQ learns the UQR to perform classification, localization and segmentation simultaneously in a regression manner. For both SQR and Mask R-CNN, full-connected layers are employed to classify the objects and regress box coordinates while the mask generated by full convolution network is supervised by 2D spatial mask.

Further, Fig. 3 shows the visualization comparison between the DETR with SQR and our SOLQ. Overall, SOLQ generates much more fine-grained masks and provides better object detection performance. We also show some failure cases in occluded environments (see Appendix A.5).

## 4.5 Ablation Studies

In this section, we also ablate several critical factors in SOLQ by progressively adjusting each factor in the system. It can be seen that each factor contributes to the final success of the SOLQ. Note that

Table 2: Comparisons between Unified Query Representation (UQR) and Separate Query Representation (SQR) on the COCO 2017 *val* set. D-DETR denotes Defoemable DETR and D-DETR* refers our reimplementment version. D-DETR* with SQR means that add an extra FPN-style branch as shown in Fig. 1(a) to perform instance segmentation on top of D-DETR*. '†' and '‡' are the results reported in ISTR [24] and Sparse RCNN [24], respectively.

| Method | Backbone | Epochs | $AP^{seg}$ | $AP^{box}$ | $AP^{box}_S$ | $AP^{box}_M$ | $AP^{box}_L$ |
|---|---|---|---|---|---|---|---|
| Faster RCNN‡ [6] | R50-FPN | 36 | - | 40.2 | 24.2 | 43.5 | 52.0 |
| Mask R-CNN† [2] | R50-FPN | 36 | 37.0 | 40.8 (+0.6) | 24.0 | 44.4 | 52.9 |
| D-DETR [35] | R50 | 50 | - | 45.4 | 26.8 | 48.3 | 61.7 |
| D-DETR* | R50 | 50 | - | 45.5 | 27.3 | 48.7 | 62.0 |
| D-DETR*+SQR | R50 | 50 | 32.2 | 45.6 (+0.1) | 27.2 | 48.8 | 61.5 |
| D-DETR*+UQR | R50 | 50 | **39.5 (+7.3)** | **47.8 (+2.3)** | **28.7** | **51.1** | **63.7** |
| Faster RCNN‡ [6] | R101-FPN | 36 | - | 42.0 | 26.6 | 45.4 | 53.4 |
| Mask R-CNN† [2] | R101-FPN | 36 | 38.8 | 43.1 (+1.1) | 25.1 | 46.0 | 54.3 |
| D-DETR* | R101 | 50 | - | 46.3 | 28.1 | 49.7 | 62.3 |
| D-DETR*+SQR | R101 | 50 | 33.4 | 46.4 (+0.1) | 28.1 | 49.9 | 62.2 |
| D-DETR*+UQR | R101 | 50 | **40.2 (+6.8)** | **48.3 (+2.0)** | **29.9** | **52.1** | **64.6** |

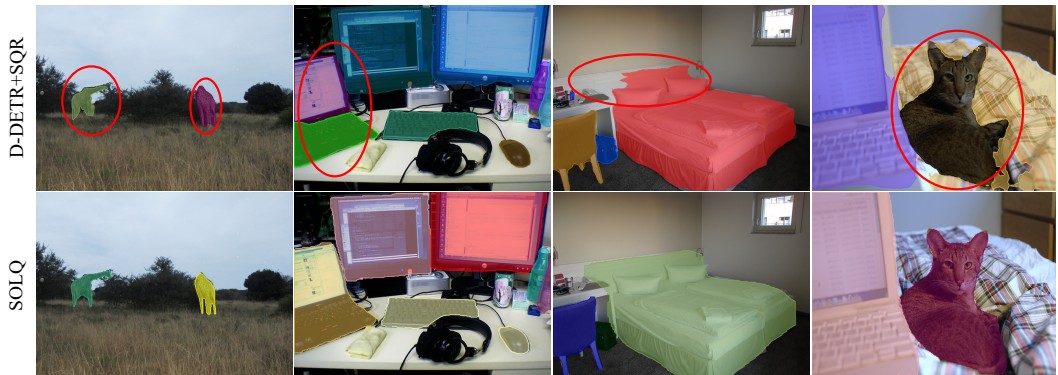

Figure 3: Visualization comparisons between Unified Query Representation (UQR) and Separate Query Representation (SQR) on the COCO 2017 *val* set. For more visualization comparison, please refer to Appendix A.4.

our ablation experiments are conducted using the single feature level with C5 and validated on the COCO 2017 *val* set.

**Mask Compression Coding Methods** We compare the impact of different compression coding methods to encode binary masks. As shown in Tab. 3a, flatting the spatial masks into 1D mask vectors directly for supervision can obtain decent segmentation result while improving $AP^{box}$ by 1%. DCT produce the best performance in both $AP^{seg}$ and $AP^{box}$ metrics. The main reason is that the loss of mask compression caused by DCT is relatively small, compared to the Sparse Coding and PCA methods. Also, DCT can be employed for each image in an online manner while Sparse Coding and PCA are performed on the whole training set to get "principal components" or "dictionary" in an offline way. Therefore, we choose DCT as our default method for mask compression coding.

**Mask Vector Loss Weight** Tab. 3b shows the effect of adjusting the weight of mask vector loss. When the weight of mask vector loss $\lambda_{vec}$ is set as 3.0, SOLQ gets the best performance in both $AP^{seg}$ and $AP^{box}$. The regression dimension of DCT vector (*e.g.* 256) is usually larger than that of the bounding box (*e.g.* 4), so the magnitude of mask vector loss is larger than that of the regression loss. Therefore, an appropriate $\lambda_{vec}$ can keep the balance between the mask branch and localization branch such that these two branches can be jointly optimized better.

**Mask Vector Loss Stage** We also ablate the impact of the number of decoder stages enabling mask vector loss in Tab. 3c. It can be seen that adding auxiliary mask vector loss on all decoders can improve by 2.9% and 1.1% on $AP^{seg}$ and $AP^{box}$, respectively. The experimental results show that auxiliary loss employed in multiple decoders is helpful to both mask and detection branches. Note

that auxiliary loss of detection branch is enabled in all decoder stages in this ablation, so the gain of detection branch is not as large as the gain of the mask branch.

**Spatial Resolution of Binary Mask** As mentioned above, mask compression coding projects the 2D $N \times N$ ground-truth binary masks into 1D mask vectors for supervision. In Tab. 3d, we explore the impact of the spatial resolution of ground truth binary mask. The segmentation performance $AP^{seg}$ is improved from 30.8 to 32.6 as $N$ increases from 64 to 128 and $AP^{seg}$ no longer improves when $N$ is greater than 128. Ground-truth binary mask with high resolution can keep more instance details but needs high-dimension mask vectors to reconstruct. Therefore, once the number of coefficients $n_k$ is chosen, $N$ has the most suitable value corresponding to $n_k$.

**Dimension of Mask Vector** Similar to the spatial resolution of ground truth binary mask, there is also a suitable value $n_k$ for given $N$. Tab. 3e shows that $n_k = 256$ achieves the best performance in both $AP^{seg}$ and $AP^{box}$ when the spatial resolution $N$ is set as 128. Theoretically, mask vector with higher dimension should reconstruct a better binary mask. However, high-dimension mask vector will enlarge the regression dimension, making it hard to optimize the mask branch. As a result, the quality of the binary mask reconstructed reduces a lot.

Table 3: Ablation studies validated on the COCO 2017 *val* set. All experiments use the single feature level with C5 in ResNet50.

(a) Impact of different binary mask encoding-decoding methods. Flatten means that reshape the 2D binary masks (28x28) into 1D mask vectors (784) directly, then optimize with L2 and dice loss jointly.

| Type | $AP^{seg}$ | $AP^{seg}_S$ | $AP^{seg}_M$ | $AP^{seg}_L$ | $AP^{box}$ | $AP^{box}_S$ | $AP^{box}_M$ | $AP^{box}_L$ |
|---|---|---|---|---|---|---|---|---|
| det | - | - | - | - | 39.4 | 20.6 | 43.0 | 55.5 |
| Flatten | 29.7 | 11.6 | 33.4 | 49.3 | 40.4 | 20.2 | 44.7 | 58.8 |
| Sparse Coding | 11.3 | 6.2 | 12.0 | 17.5 | 40.6 | 20.6 | 44.6 | 59.3 |
| PCA | 31.6 | 12.5 | 35.4 | 52.7 | 41.0 | 20.5 | 45.2 | 60.0 |
| **DCT** | **32.6** | **12.8** | **37.1** | **54.5** | **41.3** | **20.7** | **45.4** | **60.1** |

(b) Affect of adjusting mask vector loss weight. Mask vector loss is enabled only in the last decoder layer, $n_k = 256$. Spatial resolution of ground truth binary mask $N = 128$.

| $\lambda_{vec}$ | $AP^{seg}$ | $AP^{seg}_{50}$ | $AP^{seg}_{75}$ | $AP^{box}$ | $AP^{box}_{50}$ | $AP^{box}_{75}$ |
|---|---|---|---|---|---|---|
| 0.3 | 25.2 | 51.0 | 22.2 | 39.3 | 59.9 | 41.8 |
| 0.7 | 26.7 | 52.1 | 24.4 | 39.6 | 60.1 | 42.5 |
| 1 | 27.2 | 53.4 | 24.6 | 39.7 | **61.3** | **43.3** |
| 2 | 28.9 | **53.5** | 27.7 | 40.0 | 60.4 | 42.8 |
| **3** | **29.7** | **53.5** | **28.4** | **40.2** | 60.4 | 42.8 |
| 4 | 29.6 | 52.3 | 25.6 | **40.2** | 60.1 | 42.4 |

(c) Ablation of the number of decoder stages enabling mask vector loss. For example, when stage is 4, it means that enable the last 4 decoder layer with mask vector loss and $\lambda_{vec} = 3$, $n_k = 256$.

| Num. | $AP^{seg}$ | $AP^{seg}_{50}$ | $AP^{seg}_{75}$ | $AP^{box}$ | $AP^{box}_{50}$ | $AP^{box}_{75}$ |
|---|---|---|---|---|---|---|
| 1 | 29.7 | 53.5 | 27.7 | 40.2 | 60.4 | 42.8 |
| 2 | 31.8 | 55.4 | 32.1 | 40.8 | 61.2 | 43.3 |
| 3 | 32.0 | 55.2 | 32.4 | 40.7 | 61.0 | 42.9 |
| 4 | 32.4 | 55.7 | 33.1 | 41.0 | 61.2 | **43.5** |
| 5 | 32.3 | 55.3 | 33.1 | 40.9 | 60.9 | 43.4 |
| **6** | **32.6** | **55.9** | **33.4** | **41.3** | **61.7** | 43.4 |

(d) Effect of the spatial resolution of ground truth binary mask. Mask vector loss is enabled in all decoder layers and $\lambda_{vec} = 3$, $n_k = 256$.

| $N$ | $AP^{seg}$ | $AP^{seg}_{50}$ | $AP^{seg}_{75}$ | $AP^{box}$ | $AP^{box}_{50}$ | $AP^{box}_{75}$ |
|---|---|---|---|---|---|---|
| 64 | 30.8 | 54.8 | 30.8 | 40.5 | 61.2 | 42.7 |
| 96 | 31.5 | 55.4 | 31.8 | 40.7 | 61.4 | 43.1 |
| **128** | **32.6** | **55.9** | **33.4** | **41.3** | **61.7** | **43.4** |
| 256 | 32.3 | 55.4 | 32.8 | 40.6 | 60.7 | 43.3 |

(e) Impact of the dimension of mask vector. Mask vector loss is enabled in all decoder layers. $\lambda_{vec} = 3$ and spatial resolution of binary mask $N = 128$.

| $n_k$ | $AP^{seg}$ | $AP^{seg}_{50}$ | $AP^{seg}_{75}$ | $AP^{box}$ | $AP^{box}_{50}$ | $AP^{box}_{75}$ |
|---|---|---|---|---|---|---|
| 144 | 31.5 | 55.6 | 31.6 | 40.9 | 61.3 | **43.4** |
| **256** | **32.6** | **55.9** | **33.4** | **41.3** | **61.7** | **43.4** |
| 300 | 31.0 | 54.7 | 30.9 | 40.4 | 60.8 | 42.8 |
| 400 | 30.9 | 55.1 | 30.2 | 40.6 | 61.3 | 43.0 |

## 5 Conclusion

In this paper, we present SOLQ, a new instance segmentation framework. Based on DETR, SOLQ learns an unified query representation, which is used to predict the instance masks parallel with object detection in an end-to-end manner. To make the mask representation consistent with the query embedding, SOLQ projects the high-resolution spatial masks into low-dimensional mask vectors and regards mask prediction as the regression of mask vectors. SOLQ is a truly one shot framework without any two-stage operations, like ROIAlign. On the challenging COCO dataset, SOLQ achieves state-of-the-art performance on instance segmentation task and greatly improves the detection performance of DETR thanks to the multi-task learning. We believe that SOLQ will serve as a strong baseline for instance segmentation for its excellent performance and simplicity.

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
