# A   Appendix A

## A.1   Comparisons on Boundary-AP Metric

To analyze the quality of boundary prediction, we also evaluate SOLQ using the Boundary AP proposed in Boundary IoU [1] and perform the comparisons with PointRend [2] and BMask R-CNN [3] in the Tab. 1. Similar to the results on MS COCO [4], SOLQ shows much better performance on the small and medium objects while is relatively inferior on large objects. Two reasons may explain the lower performance on large objects: sparse activation of object query and fixed coding length of query. For DETR-based approaches, we observed that object queries tend to sparsely focus on specific local regions in the image, so it is relatively hard for object query to capture enough receptive field for large objects. Besides, the fixed coding length of object query also constraints the representation power for large objects. Therefore, longer/dynamic coding length of queries may be developed to adapt various sized objects. The visualization of decoder attention in A.3 also support our opinion.

Table 1: Performance comparisons under the Boundary-AP metric on COCO 2017 *val* set.

| Method | AP | $AP_S$ | $AP_M$ | $AP_L$ |
|---|---|---|---|---|
| Mask-RCNN [5] | 23.1 | 18.6 | 33.4 | 22.2 |
| PointRend [2] | **25.4** | 19.1 | 34.8 | **26.4** |
| BMask R-CNN [3] | 25.4 | 19.5 | 35.2 | 26.3 |
| SOLQ | 25.2 | **22.8** | **37.5** | 23.9 |

## A.2   Efficiency Comparisons Between SQR and UQR

As shown in Tab. 2, we further compare the number of parameters, theoretical FLOPs and FPS between the SQR and UQR. 'Mask' denotes the mask branch. UQR greatly improves the SQR performance with less parameters and computation burden.

Table 2: Comparisons on parameters, FLOPs and FPS between SQR and UQR. All models are evaluated on single Tesla V100 GPU with 512x852 input resolution.

| Method | $AP^{seg}$ | Params (M) | FLOPs (G) | FPS |
|---|---|---|---|---|
| SQR | 26.3 | 41 (D-DETR)+26.11 (Mask) | 80.13 (D-DETR)+44.88 (Mask) | 19.3 |
| UQR | 37.0 | 41 (D-DETR)+1.58 (Mask) | 80.13 (D-DETR)+0.47 (Mask)+0.0005 (iDCT) | 24.1 |

## A.3   Visualization on the Decoder Attentions

In DETR, decoder attention attends to object extremities in order to predict bounding box. Here, we visualize the decoder attention of SOLQ in Fig. 1 and it will attend to the outline of objects.

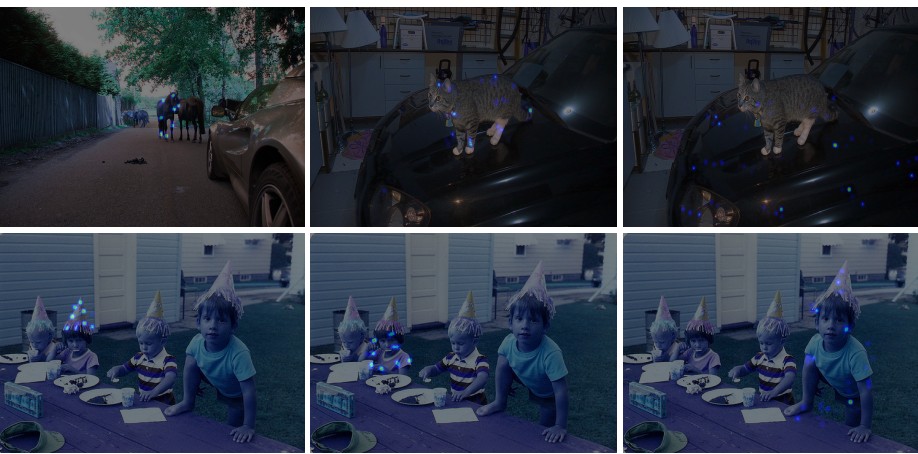

Figure 1: Visualization of decoder attentions on objects of different scales.

## A.4 More Qualitative Visualizations in Various Scenes

We add more qualitative comparisons between SQR and UQR under various situations in Fig. 2.

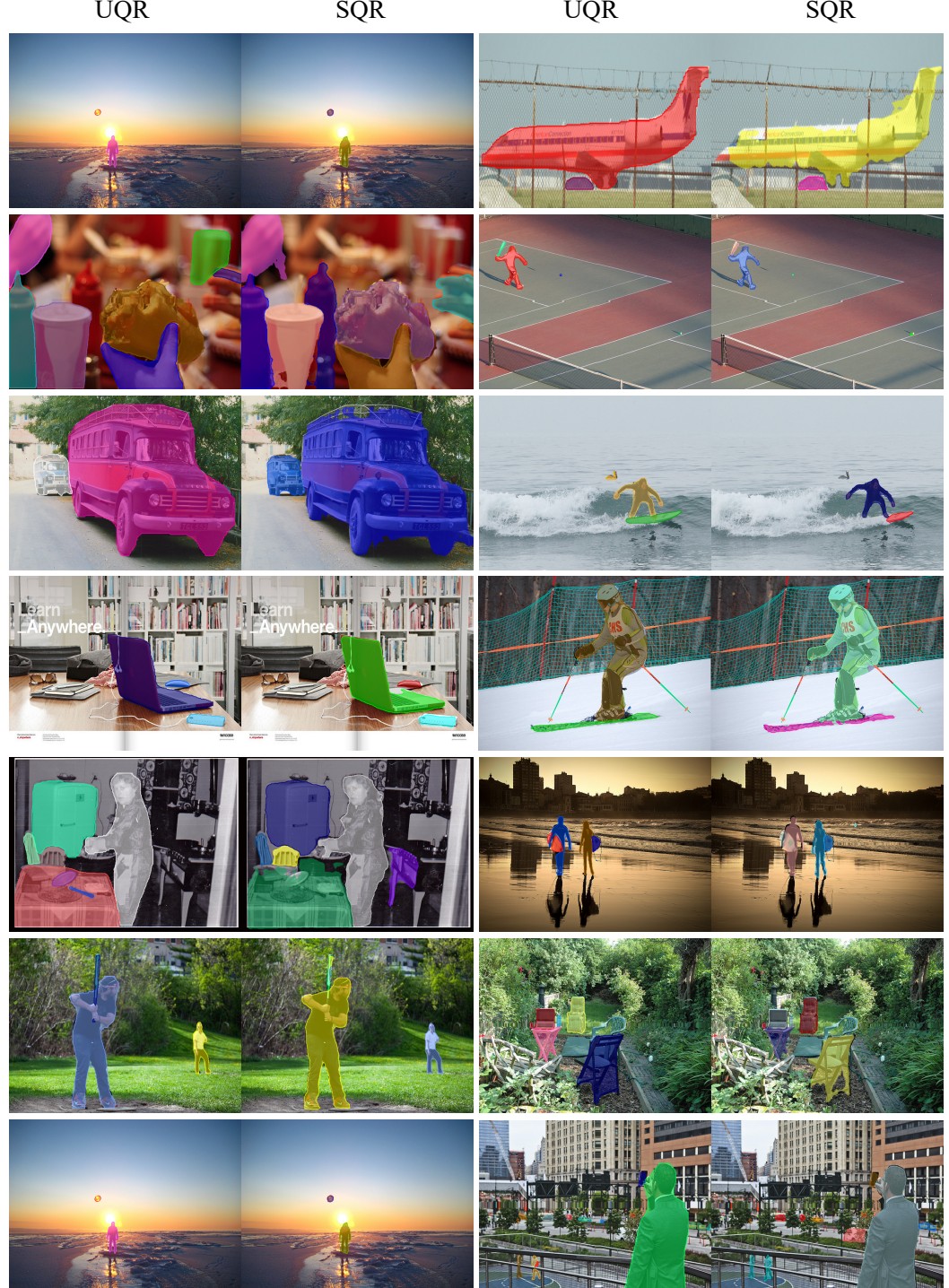

Figure 2: More visualization comparison between UQR and SQR on COCO 2017 *val* set.

## A.5  Visualizations of Failure Cases

We also observed some failure cases existing in occluded environments (see Fig. 3). Since each object query is responsible for specific region, objects with high overlap may share the same object query or correspond to adjacent object queries, resulting in the siamese mask.

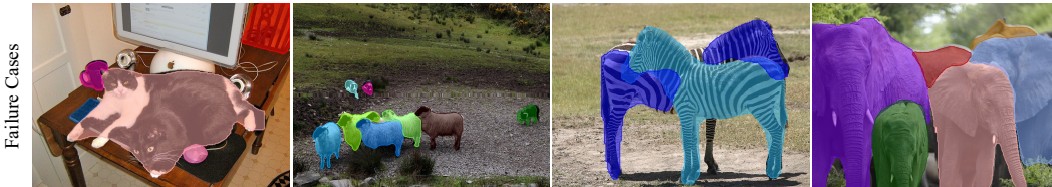

Figure 3: Visualization of some failure cases existing in occluded scenes on COCO 2017 *val* set.