# OpenReview forum: "SOLQ: Segmenting Objects by Learning Queries"
_NeurIPS.cc/2021/Conference — NeurIPS 2021 Poster_

### Official Review · Reviewer_f92m · 2021-07-15

**Rating:** 3
**Confidence:** 5

**Summary:**

This paper proposes SOLQ, which combines Deformable DETR (D-DETR) and DCT for instance segmentation. The experiments verify the effectiveness of SOLQ to a certain extent.

**Limitations And Societal Impact:**

N / A

**Main Review:**

- *Originality & Novelty:* **The novelty of SOLQ is limited.** SOLQ studies PCA and DCT for mask representation, which has already been studied in MEInst and DCT-Mask. SOLQ simply verifies DCT for mask representation again on the recently proposed D-DETR.
- *Quality:*
  - **The motivation of using sparse coding for instance segmentation is not verified in the experiment**. As stated in lines `41-43` of the paper (also in MEInst), the motivation of using sparse coding (e.g., PCA or DCT) for mask representation is that "spatial mask labels results in high computation cost, and sparse coding results in more compact mask representation". But the efficiency of sparse coding (e.g., number of parameters, theoretical FLOPs, and most importantly, FPS) is not studied in the experiments. The paper should provide some quantitative results to explain the reason to use sparse coding for instance segmentation.
  - **The spatial mask representation baseline for instance segmentation (SQR) is relatively weak.** The paper uses Separate Query Representation (SQR), which is originally proposed by DETR, as a kind of spatial mask representation baseline for instance segmentation in D-DETR framework. But SQR is a raw & straightforward design for panoptic segmentation other than instance segmentation, and the results are relatively weak. Since D-DETR / SOLQ has bounding box predictions, I suggest using the box prediction to pool RoIs for mask predictions in an interleaved manner just like HTC as a spatial mask representation baseline. Or the paper can choose to make **fair** comparisons with other arts based on spatial mask representation to verify the effectiveness and efficiency of sparse coding (e.g., PCA or DCT) for mask representation.
  - **The instance segmentation performance of using DCT is not satisfactory**. At first glance the numbers reported in the paper are OK. But the performance gap between $\mathrm{AP}^{box}$ and $\mathrm{AP}^{mask}$ is too **big**, i.e., about $8$ $\mathrm{AP}$, while the gap for the Mask R-CNN family is usually less than $6$ $AP$. This indicates that this kind of mask representation is not satisfactory. The paper claims that SOLQ can improve the object detection performance, so do Mask R-CNN family and many other instance segmentation approaches to my knowledge, since mask annotations can provide additional supervision for object detection. Note that under similar object detection performance, the instance segmentation performance of SOLQ is not competitive. An HTC with $46.3$ $\mathrm{AP}^{box}$ is $1.1$ $\mathrm{AP}^{mask}$ higher than SOLQ with $47.8$ $\mathrm{AP}^{box}$. SOLQ needs to exploit a much better object detector for a reasonable instance segmentation performance.
  - **Is SOLQ end-to-end?** I think the ultimate objective of building an end-to-end system is to remove the need for many hand-designed components and heuristics. While the use of DCT in SOLQ **introducing two extra hyper-parameters**, i.e., the spatial resolution of ground truth binary mask $N$, and the dimension of mask vector $n_k$. From the ablation study in Table 3 (d) & (e), the performance of SOLQ is **very sensitive to** these two hyper-parameters. Therefore the approach adopted by SOLQ is **clearly inconsistent with** the motivation & goal of designing an end-to-end system. On the other hand, the elimination of NMS is directly inherited from the DETR family, which shouldn't be considered as a contribution of SOLQ.
- Clarity:
  - **This paper makes many unfair comparisons in the experiments.** As a cascade / iterative framework, the qualified competitors for SOLQ are not one-stage MEInst & CondInst & SOLO, etc. But Cascade Mask R-CNN and HTC, etc. However, SOLQ does not make fair comparisons with most approaches in Table 1: SOLQ uses a longer training schedule.
  - **Figure 1 is very misleading**: the block of "DConv" has a similar size as the Res Blocks, $\mathcal{F}$ & $\mathcal{H}$ (they both have several Transformer layers), which makes the FPN-style instance segmentation design (actually, it is a panoptic segmentation design) of DETR seems to be very heavy. I think the paper should report FPS and compare the efficiency with other arts **quantitatively** other than using these bells and whistles.
  - The experiments are conducted on D-DETR, but in lines `10-12` & lines `72-75` the paper claims they observe improvements on **original DETR**, which is not verified in the experiments. Also, the abstract, introduction & method parts are built on original DETR, while the experiments switch to D-DETR. Therefore many details about the setup of the D-DETR framework are missing.
- Significance: Overall, I believe SOLQ is just yet another instance segmentation model without significant contribution to the object- or instance-level recognition tasks & model design methodology.


**Time Spent Reviewing:**

8 hours

---

> ### Author Response · Authors · 2021-08-10
> **Response to R4**
>
> We thank the reviewer for the valuable comments. However, we do not agree with some of the comments which seems unfair to us. Here are the details. For the reply to "The novelty of SOLQ is limited", please refer to Q1 in the "**General response to all reviewers**".
>
> Q1: The motivation of using sparse coding for instance segmentation is not verified in the experiment.
>
> Thanks for your constructive comments. To complement our motivation, we compare the number of parameters, theoretical FLOPs, and FPS between the SQR and UQR as suggested. Please check the results reported in the table below. Note that all the numbers are reported on a single Tesla V100 GPU with input size of 512x852. UQR greatly improves the SQR performance with less parameters and computation burden.
>
> |     Method    |     AP      |     Params(M)                 |     GFLOPs                                  |     FPS     |
> |---------------|-------------|-------------------------------|---------------------------------------------|-------------|
> |     SQR       |     26.3    |     41(d-detr)+26.11(mask)    |     80.13(d-detr)+44.88(mask)               |     19.3    |
> |     UQR       |     37.0    |     41(d-detr)+1.58(mask)     |     80.13(d-detr)+0.47(mask)+0.0005(idct)    |     24.1    |
>
> Q2: The spatial mask representation baseline (SQR) for instance segmentation is relatively weak. SQR is a raw & straightforward design for panoptic segmentation other than instance segmentation. I suggest using an interleaved manner just like HTC as a spatial mask representation baseline. Or The paper can choose to make fair comparisons with other arts based on spatial mask representation to verify the effectiveness and efficiency of sparse coding for mask representation.
>
> We disagree with you that SQR is a weak baseline. As R1 pointed out that efficient and accurate instance segmentation is largely unexplored for DETR-like models, **SQR is the strongest DETR baseline we found when we started the SOLQ project. Also, SQR is not only designed for panoptic segmentation and can directly perform instance segmentation** as shown in the last sentence of Training section in DETR repository: https://github.com/facebookresearch/detr/blob/master/d2/README.md. **We feel really confusing about the suggestion that using HTC as our baseline to verify the effectiveness and efficiency of sparse coding**. Using the box prediction to pool RoIs, like HTC, for mask predictions in DETR is not related to the mask encoding of ground-truths we think. The effectiveness and efficiency of sparse coding have been verified by the comparison between the “Flatten”, “PCA” and “DCT” in Tab.3 (a). The performance of “Flatten” is the same as the one that reshapes the query embeddings to 2D features, which are further supervised by 2-D spatial masks. Our SOLQ with DCT improves the segmentation performance by **+2.9 AP**.
>
> Q3: The instance segmentation performance of using DCT is not satisfactory. The performance gap between APbox and APmask is too big. HTC with 46.3 APbox (R-101) is 1.1 APmask higher than SOLQ with 47.8 APbox (R-50).
>
> We disagree with your opinion. **We do not think that the gap between the detection and segmentation performance is a problem**. Note that SOLQ achieves state-of-the-art performance and provides a strong baseline with a simple and effective design. **It is also unfair to compare the segmentation performance of two different methods (HTC vs. SOLQ) with different backbones (ResNet-101 vs. ResNet-50)**, showing that SOLQ is “not satisfactory”. For complementary comparison of performance, here we show both the segmentation accuracy and inference speed of HTC and SOLQ in the table below. The inference time is measured on a single Tesla V100 GPU with input size 800x1333. It shows that SOLQ achieves higher speed compared to HTC method with the same segmentation accuracy, which fails to support the opinion of R4 that the performance of SOLQ is not satisfactory.
>
> |     Methods      |     APseg    |     APbox    |     FPS     |
> |------------------|--------------|--------------|-------------|
> |     HTC-R50      |     39.7     |     44.9     |     5.8     |
> |     SOLQ-R50     |     **39.7**     |     **47.8**     |     **14.9**    |
> |     HTC-R101     |     40.8     |     46.3     |     5.5     |
> |     SOLQ-R101    |     **40.9**     |     **48.7**     |     **13.4**    |
>
> Q4: The paper claims that SOLQ can improve the object detection performance, so do Mask R-CNN family and many other instance segmentation approaches.
>
> As for the effectiveness of multi-task learning, we compared the performance between Faster-RCNN and Mask-RCNN in Tab. 2. It shows that Mask-RCNN improves the detection performance of Faster-RCNN by only **0.6%** while SOLQ improves the D-DETR by **2.3%**, revealing the effectiveness of UQR.
>
> Q5: SOLQ is not end-to-end. The use of DCT in SOLQ introducing two extra hyper-parameters. From the ablation study in Table 3 (d) & (e), the performance of SOLQ is very sensitive to these two hyper-parameters.
>
> We disagree with your opinion. The definition of an end-to-end system is not related to the hyper-parameters. As acknowledged by R1-R3, **SOLQ is definitely an end-to-end instance segmentation system.** If the spatial resolution of binary mask is an extra hyper-parameter, then any instance segmentation approach will introduce such a “hyper-parameter”, e.g., the spatial resolution of binary mask. We also disagree that “the performance of SOLQ is very sensitive to these two hyper-parameters”. The settings of spatial resolution and the dimension of mask vectors provides a trade-off between the mask quality and the compression ratio. Theoretically, if the compression ratio is very larger (when n_{k} is very small), the information loss is definitely large (corresponding to lower performance). Therefore, it is appropriate to say “choosing proper settings is important” instead of directly claiming that “SOLQ is very sensitive to these two hyper-parameters”.
>
> Q6: This paper makes many unfair comparisons in the experiments. SOLQ uses a longer training schedule.
>
> We disagree with your opinion. Methods with different mechanisms require different training schedules for full convergence. For example, SOLOv2 takes much longer ‘6x’ training schedule (72 epochs) for better performance. It should be noted that Transformer-based methods tend to require relatively longer epochs compared to CNN-based counterparts, like HTC. SOLQ mainly follows the standard training schedule (50 epochs) in D-DETR.
>
> Q7: Figure 1 is very misleading.
>
> We will modify the size of DConv block in Figure 1 to avoid the misleading presentation.
>
> Q8: The experiments are conducted on D-DETR, but in lines 10-12 & lines 72-75 the paper claims they observe improvements on original DETR, which is not verified in the experiments.
>
> DETR and D-DETR have the same detection mechanism and share the key components: object query, Transformer encoder/decoder and bipartite matching. They mainly differ in the employment of deformable attention. However, original DETR requires very long training schedule (**500 epochs**) to converge. The experiments of SOLQ are conducted on D-DETR because of its faster convergency and much shorter training schedule (**50 epochs**). We also conduct the experiments on the original DETR, SOLQ can improve the detection performance by **~2.0 AP** (38.8% ->40.7%). For clarify, we will modify the text in lines 10-12 & lines 72-75 and provide more details about the setup of the D-DETR framework in Sec. 4.2. The results based on DETR will be added to revised version.

---

> > ### Comment · Reviewer_f92m · 2021-08-13
> > **Limited novelty and unfair comparisons**
> >
> > ### Response Feedback from R4
> >
> > Unfortunately, most of my main concerns haven't been resolved. Here are some crucial concerns.
> >
> > ------
> >
> > ***(1) SOLQ has limited novelty.***
> >
> > In the response, the author claims that "Note that our contribution is building such an end-to-end framework". But the end-to-end property comes from the DETR system, not the author's contribution. Moreover, the simple combination of the DETR system and DCT has no technical contribution. SOLQ is just yet another "A + B = C" work.
> >
> > ------
> >
> > ***(2) SOLQ makes unfair comparisons, in both the manuscript and the author response.***
> >
> > In the response, the author claim that SOLQ can beat HTC. But this is an **unfair** comparison: the author uses a **50** epoch trained model optimized via **AdamW & GIoU regression loss** to compare HTC trained with **only 36** epoch training optimized via **vanilla SGD w/o GIoU regression loss**. As shown in the Swin Transformer [1], the R-CNN system can also benefit from AdamW & GIoU loss, etc.
> >
> > Here is a comparison between Cascade Mask R-CNN and SOLQ under relative fair settings. The Cascade R-CNN is optimized using AdamW & GIoU regression loss, which is same as DETR & SOLQ.
> >
> > | Method                            | Training Schedule          | Mask AP         | FPS             |
> > | --------------------------------- | -------------------------- | --------------- | --------------- |
> > | Cascade Mask R-CNN, ResNet-50 [1] | 36 epochs                  | 40.1            | 18.0            |
> > | SOLQ, ResNet-50                   | 50 epochs (**+14** epochs) | 39.7 (**-0.4**) | 14.9 (**-3.1**) |
> >
> > [1] "Swin Transformer: Hierarchical Vision Transformer using Shifted Windows". In ICCV 2021.
> >
> > **Even a Cascade Mask R-CNN  with a shorter schedule can beat SOLQ in terms of both AP and FPS.** I believe HTC will crush SOLQ under fair comparisons.
> >
> > Moreover, SOLQ doesn't perform ***speed (both training and Inference) comparisons*** with other methods in the manuscript. In the rebuttal, the authors claim that it obtains better APs than SOLOv2. However, SOLOv2 is designed for fast instance segmentation and it obtains 37.1% AP with 31.3 FPS. I would like to emphasize that the paper should make fair comparisons.

---

> > > ### Author Response · Authors · 2021-08-13
> > > **Respond to R4**
> > >
> > > We thank R4 for the comments.
> > >
> > > Q1: SOLQ has limited novelty. The simple combination of the DETR system and DCT has no technical contribution. SOLQ is just yet another "A + B = C" work.
> > >
> > > We feel that “SOLQ is just yet another "A + B = C" work” is rather unfair for us. DETR is a popular framework, which is worthy to be further explored in depth. However, it is still underexplored that building an end-to-end instance segmentation framework based on DETR. There are several challenges when adapting the DETR to instance segmentation. Since the DETR is developed based on 1-D object query while the instance mask is 2-D, such inconsistency in domain will hinder the whole framework learning from accurate spatial information. Therefore, we consider encoding the instance information into object queries. Mask encoding approaches work well on embedding instance information into query vectors. We believe that both the analysis on domain inconsistency and adapting the mask embedding into DETR are all our contribution.
> > >
> > > Furthermore, when R4 compared SOLQ with DCT and PCA, it is easy to find the motivation is very different. MEInst based on a fully convolutional network regresses the PCA components while DCT-Mask built on Mask-RCNN crops the RoI features by RoIAlign to predict the DCT coefficients. Both of them focus on compressing the instance masks to reduce the computation cost in one-stage or two-stage objectors. SOLQ first concludes the reason behind pool performance on instance segmentation is domain inconsistency between the object query and instance mask; then proposes to adapt the mask encoding approaches into DETR to achieve the consistency between them.
> > >
> > > There are many other choices applying MEInst or DCT-Mask into DETR. For example, one can follow MEInst to employ the fully convolutional network (FPN style in DETR) to predict the PCA components. One can also follow DCT-Mask, performing RoIAlign to crop RoI features for mask encoding. It is obvious that we find a favorable and non-trival solution. We suggest R4 consider more about our contribution on motivation and analysis, i.e., **how to use DETR on segmentation task in an effective and elegant way**, rather than only focus on the eventual methodology.
> > >
> > > Q2: SOLQ makes unfair comparisons, in both the manuscript and the author response.
> > >
> > > We feel the comparison you listed between the Cascade Mask-RCNN and our SOLQ is still unfair. Note that the result of Cascade Mask-RCNN with ResNet-50 is from Swin Transformer where the Sync-BN is employed in RoI head, whose effect is non-negligible. Please refer to here https://github.com/SwinTransformer/Swin-Transformer-Object-Detection/blob/master/configs/swin/cascade_mask_rcnn_swin_tiny_patch4_window7_mstrain_480-800_giou_4conv1f_adamw_3x_coco.py#L36 for more details. It should be noted again that Transformer-based methods tend to require relatively longer epochs compared to CNN-based counterparts. Also, it is known that AdamW optimizer is more suitable than SGD for Transformer-based architectures. Here we further provide the comparison between the SOLOv2 and SOLQ in the table below. The inference speed is measured following the same condition as in the official SOLOv2 repository and paper with input size 512x852. The official repository of SOLOv2 is at https://github.com/WXinlong/SOLO/blob/master/README.md. The results show that SOLQ achieves higher segmentation performance and SOLOv2 is a bit faster than SOLQ under the same condition.
> > >
> > > |     Methods       |     Note                   |     APseg    |     FPS     |
> > > |-------------------|----------------------------|--------------|-------------|
> > > |     SOLOv2-R50    |     Reported   in paper    |     37.1     |     31.3    |
> > > |     SOLOv2-R50    |     Reported   in repo     |     36.4     |     29.4    |
> > > |     SOLQ-R50      |     /                      |     37.4     |     24.1    |

---

> > > ### Comment · Reviewer_xtMD · 2021-08-15
> > > **Novelty is not all you need**
> > >
> > > I feel like novelty is quite a subjective quantity, and not necessarily useful to measure a paper’s worth. Simple methods often scale much better than those with many bells and whistles, and simplicity is deceivingly hard. HTC for example, is basically Cascade R-CNN with a very minor modification to the order of operations, and yet is still a strong baseline two years after it was initially proposed.
> > > Every research is a combination of existing ideas. If you want to go that way, you may as well call DETR a "A + B = C method", as it is a mere combination of transformers+hungarian loss (none of which were introduced in the paper). Of course, that hides the fact that there are a lot of details that needs to be addressed behind the scenes to make it work in practice, and that hindsight is always 20/20.
> > >
> > > For this particular work, the claim stated at the end of the abstract is "We hope our SOLQ can serve as a strong baseline for the Transformer-based instance segmentation". This claim doesn’t involve being the fastest nor the strongest detector in the world. It involves proposing a strong *fully differentiable* instance segmentation model.
> > > Let’s pick this claim apart:
> > > - How do we know it is *strong* ?: well, it definitely performs in a similar ballpark as established baselines. Sure, as you pointed out, one can make the baselines stronger. The performance of any method improves over time, not because the underlying method changes, but because in years of existence the community finds better hyper-parameters and training recipes. The same will likely happen for SOLQ. (BTW I checked the SWIN paper, and they don’t provide much details on the training details for their Cascade results, so one has to rely on digging in the code, not ideal).  Exact apple-to-apple are almost impossible when the systems are that different.
> > > - What about the fully differentiable part? Well *some* of it is inherited from DETR, as you pointed out. However, DETR’s mask head is too computationally demanding, so the authors had to resort to a two stage training where they first train the detection part, freeze it, then train the mask head, meaning it’s not really end-to-end. So in that way, in my view, SOLQ is *the first fully end-to-end instance segmentation model*. Moreover, they provide a convincing apple-to-apple comparison with the closest model, which is DETR’s "almost end-to-end" mask head and show increased performance across the board (and, I find it much healthier that the mask head helps SOLQ’s box AP, which was not the case for DETR).
> > >
> > >
> > > Why would we care about an end-to-end instance segmentation model? Well, being end-to-end means that it can be plugged in downstream applications that require segmentation, while still retaining the possibility to differentiate through the whole pipeline. Several systems requiring detection and built upon DETR have already been proposed:
> > > - For tracking "TrackFormer: Multi-Object Tracking with Transformers", Meinhardt et al
> > > - For text+vision (eg VQA) "MDETR: Modulated Detection for End-to-End Multi-Modal Understanding", Kamath et al
> > > Being able to add segmentation in an end-to-end way into these pipelines is an exciting perspective, one that was not possible before this paper, and certainly not with the Mask-RCNN/HTC family.
> > >
> > >
> > > At the end of the day, I would encourage the other reviewers to keep the big picture in mind, and not become too focused on absolute performance and SOTA chasing, which this paper never claimed to achieve in the first place.

---

### Official Review · Reviewer_94Kv · 2021-07-16

**Rating:** 5
**Confidence:** 5

**Summary:**

This paper tries to combine DETR and Mask Encoding techniques for end-to-end instance segmentation: DETR could directly separate different objects and Mask Encoding could encode the arbitrary instance mask into a low-dimensional and fix representation. To verify the effectiveness of each component, the authors conducted extensive ablations for each component. The final performance is inspiring, espically using Trasnformer backbone.

**Ethics Review Area:**

["I don’t know"]

**Main Review:**

== Pros ==

1. The paper is well motivated and shows a reasonable story. The wring is good and easy to follow.
2. Using Mask Encoding to encode the mask is a good choice for modeling the mask, and the performance (especially AP_s) has been largely improved.
3. The UQR is better than SQR, and it could further improve the object detection performance.

== Cons ==

1. My main concern is originality. From my understanding, Mask Encoding/Compression has been explored by previous work (i.e., DCT-Mask, MEInst). So adopting it into DETR is straightforward. It could only demonstrate that Mask Encoding works well for DETR-style architecture, but both techniques are not your contribution.
2. I agree that no previous work has tried combing these techniques together to form an end-to-end instance segmentation system. The performance is also great. But combining weakened your contribution.
3. In L207, the authors claim that DCT-Mask is not learned in an end-to-end manner. However, I believe DCT-Mask is in fact an end-to-end trainable system (please refer to Sec.3 of their paper).
4. SOLQ gets higher AP on small objects, however, it suffers from large objects compared with SOLOv2 (Table 1). Could you explain why?
5. What is the difference between UQR and SQR? From my understanding, UQR is using the encoded mask for supervision, while SQR is using a binary mask.

== Overall ==

Including Mask Encoding into DETR could directly enable DETR to do instance segmentation. That is good. However, Mask Encoding has been used in the previous two-stage (DCT-Mask), and one-stage (MEInst) methods.
If you just change the detector from Faster R-CNN/FCOS to DETR, the novelty would be really incremental. It is
important to discuss the differences and highlight the novelty.



**Time Spent Reviewing:**

3 hours

---

> ### Author Response · Authors · 2021-08-10
> **Response to R3**
>
> We thank R3 for the careful review and the overall positive feedback.
>
> Q1: In L207, the authors claim that DCT-Mask is not learned in an end-to-end manner. However, I believe DCT-Mask is in fact an end-to-end trainable system.
>
> We regret for the wrong statement. DCT-Mask is indeed an end-to-end trainable system. We will revise the statement here.
>
> Q2: SOLQ gets higher AP on small objects, however, it suffers from large objects compared with SOLOv2 (Table 1). Could you explain why?
>
> Lower performance on large objects has two reasons: **sparse activation of object query** and **fixed coding length of query**. For DETR-based approaches, we observed that object queries tend to sparsely focus on specific local regions in the image, so it is relatively hard for object query to capture enough receptive field for large objects. As discussed in R1-Q5, the fixed coding length of object query also constraints the representation power for large objects. Therefore, longer/dynamic coding length of queries maybe developed to adapt various sized objects. We will add more discussion regarding to the performance gap between small and large sized objects.
>
> Q3: What is the difference between UQR and SQR? From my understanding, UQR is using the encoded mask for supervision, while SQR is using a binary mask.
>
> In the view of supervision signal, we agree with you that UQR uses the encoded mask for supervision while SQR uses the binary mask. However, they are significantly **different in the representation learning**. For SQR, query embeddings are directly reshaped to 2-D spatial domain, which is not consistent with the detection branch. **Such inconsistency in domain will hinder the whole framework benefiting from multi-task learning**. As described in the introduction, both the Transformer encoder and decoder fail to model the spatial information well. It is inappropriate to generate the spatial mask based on 1-D query embeddings. While for the UQR, it aims to **encode the instance segmentation information into the 1-D query embeddings, achieving parallel end-to-end prediction**. All predictions are obtained in a regression manner.

---

> > ### Comment · Reviewer_94Kv · 2021-08-13
> > **Thanks for the response, however my main concern has not been resolved**
> >
> > My main concern of this paper is originality (please refer to my question 1 and 2). Unfortunately, it has been ignored by the authors' response.
> >
> > In my initial comment, I suggest the authors discuss the differences SOLQ and previous works (MEInst and DCT-Mask).  In MEInst the binary masks are represented by offline PCA-based encoding, while in DCT-Mask, they are encoded by online DCT-transformation.
> > From my understanding, you are trying to combine DETR with Mask Ecoding process, and the main difference is the DETR-style detector.
> >
> > However, both DETR detector or  Mask Encoding are not originally proposed by SOLQ. If the authors could not tell the differences and highlight the novelty, I'm not able to give positive feedback.

---

> > > ### Author Response · Authors · 2021-08-13
> > > **Response to R3**
> > >
> > > About the novelty issue, please kindly refer to our reply to Q1 in the “**General response to all reviewers**”. Briefly speaking, introducing mask encoding into our framework is non-trivial. Both the purpose and approach are very different with existing works, like MEInst and DCT-Mask. MEInst and DCT-Mask aim to compress the instance masks to reduce the computation cost in one-stage or two-stage objectors. In our framework, **the key idea is to embed instance information into query vectors for better multi-task learning**. However, the instance masks are 2-D so we seek for an encoding method to compress the segmentation mask ignoring the spatial information. We find the existing methods, such as DCT, can achieve it. It does not mean that we just combine it with DETR detector. **There are many choices applying MEInst or DCT-Mask into DETR out of our motivation. If we directly follow the way in DCT-Mask, we can integrate the head of DCT-Mask (performing RoIAlign to crop RoI features for mask encoding) on top of DETR. Clearly, it is not as elegant as our method**.

---

> > > > ### Comment · Reviewer_94Kv · 2021-08-13
> > > > **About the originality response**
> > > >
> > > > Generally, I agree with R4 that the novelty is limited.
> > > >
> > > > --The end2end property comes from the **set prediction** which was previous proposed by DETR, thus it is not the paper's contribution.
> > > >
> > > > --I believe we must show respect to the previous works, especially those that inspired ours.
> > > > Since instance information embedding is originally proposed by MEInst (offline) and DCT-Mask (online), you should give an in-depth comparison in the main section of the paper, not just briefly mentioned them in the related work (L106).
> > > >
> > > > --If the motivation for this work is better embedding-based multi-task learning, I think the authors must point out this goal in the intro/method section. More importantly, the authors should give a more in-depth analysis of how and why embedding-based learning is better.

---

> > > > > ### Author Response · Authors · 2021-08-13
> > > > > **Respond to R3**
> > > > >
> > > > > We thank R3 for the constructive comments.
> > > > >
> > > > > Q1: The novelty is limited.
> > > > >
> > > > > For clarification on the novelty, please kindly refer to our latest response to R4-Q1 at https://openreview.net/forum?id=78GFU9e56Dq&noteId=hS6cCLTdkiC.
> > > > >
> > > > > Q2: I believe we must show respect to the previous works, especially those that inspired ours. Since instance information embedding is originally proposed by MEInst (offline) and DCT-Mask (online), you should give an in-depth comparison in the main section of the paper, not just briefly mentioned them in the related work (L106).
> > > > >
> > > > > Thanks for your kind reminder. We have compared the differences with MEInst and DCT-Mask in the Sec. 3.3 in detail, not just briefly mentioned them in the related work. Performance comparisons with them are also conducted in the experiment section. We think we have shown enough respect to the previous works. For more comparisons with MEInst and DCT-Mask, please also refer to our latest response to R4-Q1.

---

### Official Review · Reviewer_AEBz · 2021-07-18

**Rating:** 9
**Confidence:** 5

**Summary:**

The paper proposes and end-to-end instance segmentation approach based on recently proposed Transformer-based object detector DETR. In DETR instance segmentation is done by upsampling of ResNet features from multiple stages, which is computationally expensive. Instead, the authors propose to predict instance masks together with bounding box and class labels, by compressing the masks. This results in a simple and efficient detection and instance segmentation model. Moreover, AP on both tasks improves significantly over DETR and Deformable DETR, achieving very impressive results on par with state-of-the-art, on the challenging COCO dataset. The paper also provides a detailed ablation study of the proposed model.

**Limitations And Societal Impact:**

The authors addressed the limitations and potential negative societal impact of their work adequately.

**Main Review:**

Originality: the paper is based on an original and novel idea, predicting compressed instance masks directly with DETR model, avoiding complex operations like proposal generation, non-maximum suppression and ROIAlign, compared to Mask R-CNN.

Quality: the proposed approach to instance segmentation is simple and elegant. The experimental evaluation fully supports the claims, and there is a detailed ablation study for hyperparameter and representation choices. The performance of the proposed approach on COCO benchmarks is very impressive. Similarly to Mask R-CNN Vs Faster R-CNN, adding mask supervision improves bounding box prediction performance.

Clarity: the paper is very well written and is easy to follow. I have only a few questions/suggestions to improve the paper:
- attention visualizations in DETR show that decoder attends to object extremities in order to predict bounding box. It would be interesing to see SOLQ decoder attention visualizations, if it attends to object outline?
- I found "vector loss" confusing, would "Mask vector loss" be a better name?
- would be great to have more details on the Swin-SOLQ experiments, is it "almost" CNN-free object detector?
- I did not find if L_vec loss is also used in cost for bipartite matching?

Significance: the paper greatly improves over DETR and Deformable in detection and segmentation performance, and computational complexity. I expect it to be become a new baseline for both tasks.

**Time Spent Reviewing:**

5

---

> ### Author Response · Authors · 2021-08-10
> **Response to R2**
>
> We thank R2 for the positive comments and valuable feedback. For the reply to "Swin-SOLQ experiments" and "if L_vec loss is also used in cost for bipartite matching", please refer to Q2 and Q3 in the "**General response to all reviewers**".
>
> Q1: Attention visualizations in DETR show that decoder attends to object extremities in order to predict bounding box. It would be interesting to see SOLQ decoder attention visualizations, if it attends to object outline?
>
> Thanks for your valuable suggestion. We have performed the visualization on the decoder attention of SOLQ as suggested and it will attend to the outline of objects. Please refer to the visualization examples at https://github.com/Anonymous1406/SOLQ/tree/main/vis_attns.
>
> Q2: I found "vector loss" confusing, would "Mask vector loss" be a better name?
>
> Thanks for your comments. We will revise accordingly.

---

### Official Review · Reviewer_xtMD · 2021-08-02

**Rating:** 7
**Confidence:** 5

**Summary:**

The paper proposes a different approach to instance segmentation using DETR-like
architecture. Instead of the FPN-based approach adopted in the original DETR, they opt for
regressing directly an encoded version of the mask.

This approach allows to perform segmentation in a cheaper manner, and also allows to use the
segmentation objective in the auxiliary loss (not possible in original DETR due to
prohibitive cost), which gives some substantial improvement.

The results shown on Coco are competitive with current state of the art.


**Limitations And Societal Impact:**

Nothing stands out.

**Main Review:**

The method is well motivated, and the paper is easy to follow. The ablations thoroughly
explore the technical choices being made (although I have some questions about the said
ablations, see below).

Overall, I find the contribution valuable for the community, since efficient and accurate
instance segmentation is largely unexplored for DETR-like models. I thus lean to accept this
paper. In the rest of this review, I list some questions and possible ways to make the
submission even stronger.

Note that [4] also tackled instance segmentation with a DETR-like architecture (although
mainly focusing of video), and should be discussed in the related work section.

## Comparison with state-of-the-art - SWIN results

I am a bit confused by the presence of the SWIN-L backbone in the main table, without much
explanations.

While it is interesting in theory to combine DETR and SWIN (that creates the first
"transformer only" detector that I am aware of), proper comparison should be made with other
systems. The SWIN paper itself provides numerous detection results, including a 51.1 mask
AP, much better than what is being reported in this paper. It is to be noted that direct
comparison is a bit difficult since this particular result involves numerous bells and whistles. I
do think, however, that a proper comparison, on comparable settings, is advisable for
detection results using this backbone.


## Ablations consistency

From table 3.b, it appears that the optimal coefficient for the $\lambda_\text{vec}$
parameter is 3.0. However, ablations in 3.c, 3.d and 3.e are all conducted with $\lambda_\text{vec}=0.3$

What is the reason behind this? I am not sure what to conclude from ablations performed with
sub-optimal hyper-parameters.

## Timings

Training time could be reported for completeness.

More importantly, I believe it would be valuable to provide inference timings for this
model, compared to the competing ones. One of the major claim of the paper is the simplicity
(one-shot, no ROIalign, ...), but it is not clear that the simplicity actually translates to
faster inference.
In particular, it is not clear to me how costly the DCT decoding actually is.
Similarly, if possible a FLOPS analysis would be interesting.


## Mask quality
Recent papers such as [1] have argued that Mask-IoU doesn't capture the full picture in the
quality of the masks. For example, despite reaching fairly good AP, it is known that
Mask-RCNN masks are sometimes "blobby". It would be interesting to provide more qualitative
comparisons of the performance of the model in various situations. Fig. 4 partially answers
that question by showing failure modes in occluded objects, but what about, for example,
high frequency details in the mask boundaries? (eg the individual fingers of a hand) Would
they be captured adequately?

Beyond qualitative results, it would also be interesting to provide quantitative metrics,
such as the Boundary-IoU [1], and compare with recent high-quality mask prediction models
such as PointRend [2] and BMask R-CNN[3]

## Segmentation loss in matching?

From the equations provided in section 3.1 and 3.2, it would seem that the segmentation loss
isn't used as part of the matching. Is this correct?
If so, have you also tried incorporating it in the matching cost? Intuitively, it is
desirable to make the loss and the matching as similar as possible, to ensure a smoother
optimization.


## Panoptic segmentation?

In the original DETR paper, one of the main application of the segmentation head was
Panoptic segmentation. How easy would it be to apply this method to this setting as well? In
particular, does the mask encoding work well for stuff segments? Would it be easy to merge
the predictions from different masks? (similar to the pixel-wise argmax done in DETR).



[1] Boundary IoU: Improving Object-Centric Image Segmentation Evaluation, Cheng et al

[2] PointRend: Image segmentation as rendering, Kirillov et al

[3] Boundary-preserving Mask R-CNN, Cheng et al

[4] End-to-End Video Instance Segmentation with Transformers, Wang et al

# Post rebuttal
In light of the rebuttal, I’m increasing my score (6->7)

**Time Spent Reviewing:**

6

---

> ### Author Response · Authors · 2021-08-10
> **Response to R1**
>
> We thank the reviewer for the positive comments and valuable feedback. For the reply to "Comparison with state-of-the-art - SWIN results" and "Segmentation loss in matching", please refer to Q2 and Q3 in the "**General response to all reviewers**".
>
> Q1: Note that VisTR [1] also tackled instance segmentation with a DETR-like architecture and should be discussed in the related work section.
>
> Thanks for your good suggestion. We agree with that and will cite and discuss it in our related work.
>
> Q2: From table 3.b, it appears that the optimal coefficient for the λ parameter is 3.0. However, ablations in 3.c, 3.d and 3.e are all conducted with 0.3.
>
> Thanks for your pointing out. It is a typo. We have revised and will modify the caption of Tab. 3 to keep the ablations consistency. The optimal λ parameter is 3.0 and it is conducted in the rest of ablations.
>
> Q3: Training and inference time could be reported for completeness. Similarly, if possible, a FLOPS analysis of DCT decoding would be interesting.
>
> Thanks for your good suggestions. For completeness, we report the training and inference time in the table below. The FLOPS of DCT decoding is also summarized in it. The inference speed is measured on a single Tesla V100 GPU with input size 512x852. For efficiency comparison between SQR and UQR, please kindly refer to the R4-Q1.
>
> |     Training time            |     Inference speed    |     Flops of DCT decoding    |
> |------------------------------|------------------------|------------------------------|
> |     ~48 Hours [50 epochs]    |     24.1 FPS           |     0.51 B                   |
>
> Q4: It would be interesting to provide more qualitative comparisons of the performance of the model in various situations.
>
> Thanks for your constructive suggestions. We have uploaded more qualitative comparisons in various situations to https://github.com/Anonymous1406/SOLQ/tree/main/vis_figs.
>
> Q5: It would be interesting to provide quantitative metrics, such as Boundary-IoU [2], and compare with recent high-quality mask prediction models, such as PointRend [3] and BMask R-CNN [4].
>
> Thanks for your kind reminder. We evaluate SOLQ under the Boundary-IoU as suggested and perform the comparison with PointRend and BMask R-CNN in the Table below. Similar to the results on MS COCO, SOLQ shows much better performance on the small and medium objects and is relatively inferior on large objects. In SOLQ, we use fixed coding length (e.g., 256) so we feel the results are reasonable. For fair comparison, we should also compare the results with Mask-RCNN (23.9% vs. 22.2% on APl), which is also based on fixed length representation. We believe that longer/dynamic coding length, may make up the information losses for large objects. We will cite these three papers, add performance comparison and discussion under the Boundary-IoU metric. Please also refer to the R3-Q2 for more analysis on the performance variances on different objects.
>
> |     Methods          |     AP      |     APs     |     APm     |     APl     |
> |----------------------|-------------|-------------|-------------|-------------|
> |     Mask-RCNN        |     23.1    |     18.6    |     33.4    |     22.2    |
> |     PointRend        |     25.4    |     19.1    |     34.8    |     **26.4**    |
> |     BMask   R-CNN    |     25.4    |     19.5    |     35.2    |     26.3    |
> |     SOLQ             |     25.2    |     **22.8**    |     **37.5**    |     23.9    |
>
> Q6: How easy would it be to apply this method to panoptic segmentation? In particular, the mask encoding work well for stuff segments? Would it be easy to merge the predictions from different masks?
>
> SOLQ may fail to directly work for stuff segments, which are usually of very large size. It will result in large information loss when compressing them into low-dimension vectors by mask encoding. One can encode the stuff segments by dividing stuff segments into several patches and conduct mask encoding for each patch. It works well for us on pure semantic segmentation task (For example, mIoU=77.73% on Cityscapes dataset with Swin Tiny backbone). In this way, we can simply merge the predictions from different masks.
>
> [1] End-to-End Video Instance Segmentation with Transformers, Wang et al
> [2] Boundary IoU: Improving Object-Centric Image Segmentation Evaluation, Cheng et al
> [3] PointRend: Image segmentation as rendering, Kirillov et al
> [4] Boundary-preserving Mask R-CNN, Cheng et al

---

> > ### Comment · Reviewer_xtMD · 2021-09-02
> > **Rebuttal acknowledgement**
> >
> > I thank the authors for their thorough rebuttal.
> >
> > I am pleased to see that the proposed method compares positively with competing approaches when evaluated using the BoundaryIOU metric. I believe adding these results in the paper (or appendix) would make the case stronger.
> >
> > Similarly, some of the visualization provided are interested and could be added in appendix.
> >
> >
> > In light of the rebuttal, I am increasing my initial rating (6->7)

---

### Author Response · Authors · 2021-08-10
**General response to all reviewers**

We thank the reviewers for the evaluation of our manuscript for the positive feedback: valuable contribution and novel idea (R1, R2), well-motivated and written (R1, R2, R3), simple and elegant approach (R2), thorough and detailed ablations (R1, R2), impressive performance (R2, R3). For reproduction, code is anonymously released at https://github.com/Anonymous1406/SOLQ.

Q1(R3, R4): The novelty of SOLQ is limited. SOLQ studies PCA and DCT for mask representation, which has already been studied in MEInst and DCT-Mask.

Our contribution: **SOLQ encodes the instance segmentation information into the query vectors based on DETR framework, thus enabling end-to-end prediction**. Since query vectors are 1-D while the instance masks are 2-D, so we need to find an encoding approach to compress the masks into 1-D vectors. We try some compression encoding methods in literatures. Note that our contribution is building such an end-to-end framework. **Mask encoding approaches, such as PCA and DCT, are not our contribution and not claimed to be the novelty** in the manuscript. They simply provide some choices that enables the mask branch learned in a regression manner.

Q2(R1, R2): I am a bit confused by the presence of the SWIN-L backbone in the main table, without much explanations. A proper comparison, on comparable settings, is advisable for detection results using this backbone.

We thank both R1 and R2 for the valuable suggestion. As R1/2 point out, the combination of Swin Transformer and SOLQ formulate a "**transformer only**" detector without any CNN components. We will add more description and provide more background about why developing the experiments with Swin-Transformer. For fair comparison, we conduct the experiment of SOLOv2 with Swin-L backbone. Experimental results show that SOLQ outperforms SOLOv2 by **+0.6%** on COCO val set. More Swin-based results shall be added to our project.

Q3(R1, R2): Whether the mask loss is used in the cost for bipartite matching?

Thanks for your pointing out this problem. For the results reported in the submission version, the segmentation branch is not included in bipartite matching. We further conduct the experiments using the mask loss as part of bipartite matching and the results show inferior performance (-0.7%) than those ignoring mask loss in bipartite matching. Large loss jitter is observed during the initial training stage. The participation of mask loss may affect the global matching between the object queries and ground-truths. We also notice that mask loss is not included in bipartite matching in VisTR [1].

[1] End-to-End Video Instance Segmentation with Transformers, Wang et al

---

### Decision · Program_Chairs · 2021-09-27

**Decision:**

Accept (Poster)

**Comment:**

The paper presents a direct way to do instance segmentation in DETR: instead of producing segmentation masks with an FPN, it regresses segmentation masks from compressed representations as queries. This allows training DETR for segmentation end-to-end (while the original DETR and follow-ups have segmentation trained as a second step to detection).

This paper is roughly the result of the combination of (DF-)DETR with mask encoding (DCT), which is criticized by reviewer 94Kv and f92m. I believe there is nonetheless originality in making it work so well that the approach reaches very strong numbers in segmentation on COCO (39.7 APseg with a ResNet-50 backbone and 45.9 with a Swin-L).

Reviewer f92m constructively points out that the gap between APbox and APseg is bigger for this approach than for others, which could mean that it requires a stronger detector. Another interpretation can be that training the DF-DETR model with this additional segmentation loss end-to-end boosts the APbox.

Overall, I believe the paper presents a significant contribution, and the authors answered some of the concerns of the reviewers. This is suitable for publication at NeurIPS.